# Map1lc3b and Sqstm1 Modulated Autophagy for Tumorigenesis and Prognosis in Certain Subsites of Oral Squamous Cell Carcinoma

**DOI:** 10.3390/jcm7120478

**Published:** 2018-11-24

**Authors:** Pei-Feng Liu, Hsueh-Wei Chang, Jin-Shiung Cheng, Huai-Pao Lee, Ching-Yu Yen, Wei-Lun Tsai, Jiin-Tsuey Cheng, Yi-Jing Li, Wei-Chieh Huang, Cheng-Hsin Lee, Luo-Pin Ger, Chih-Wen Shu

**Affiliations:** 1Department of Medical Education and Research, Kaohsiung Veterans General Hospital, Kaohsiung 81362, Taiwan; d908203@gmail.com (P.-F.L.); angioadsc@gmail.com (C.-H.L.); lpger@isca.vghks.gov.tw (L.-P.G.); 2Department of Optometry, Shu-Zen Junior College of Medicine and Management, Kaohsiung 82144, Taiwan; 3Cancer Center, Kaohsiung Medical University Hospital, Kaohsiung Medical University, Kaohsiung 80708, Taiwan; changhw2007@gmail.com; 4Department of Biomedical Science and Environmental Biology, Kaohsiung Medical University, Kaohsiung 80708, Taiwan; 5Department of Internal Medicine, Kaohsiung Veterans General Hospital, Kaohsiung 81362, Taiwan; jcheng@vghks.gov.tw (J.-S.C.); tsaiwl@yahoo.com.tw (W.-L.T.); 6Department of Pathology and Laboratory Medicine, Kaohsiung Veterans General Hospital, Kaohsiung 81362, Taiwan; hplee0627@vghks.gov.tw; 7Department of Nursing, Meiho University, Pingtung 91202, Taiwan; 8Oral and Maxillofacial Surgery Section, Chi Mei Medical Center, Tainan 71004, Taiwan; ycysmc@gmail.com; 9Department of Dentistry, Taipei Medical University, Taipei 11031, Taiwan; 10School of Medicine, National Yang-Ming University, Taipei 11221, Taiwan; 11Department of Biological Science, National Sun Yat-sen University, Kaohsiung 80424, Taiwan; tusya@faculty.nsysu.edu.tw (J.-T.C.); lee720127@yahoo.com.tw (Y.-J.L.); 12Graduate Institute of Integrated Medicine, China Medical University, Taichung 40402, Taiwan; jeff20628@gmail.com; 13School of Medicine for International Students, I-Shou University, Kaohsiung 82445, Taiwan; 14Institute of Biomedical Sciences, National Sun Yat-sen University, Kaohsiung 80424, Taiwan

**Keywords:** MAP1LC3B, SQSTM1, autophagy, subsites, tumorigenesis, prognosis, oral cancer

## Abstract

Oral squamous cell carcinoma (OSCC) is one of the most common cancer types worldwide and can be divided into three major subsites: buccal mucosal SCC (BMSCC), tongue SCC (TSCC), and lip SCC (LSCC). The autophagy marker microtubule-associated protein light chain 3B (MAP1LC3B) and adaptor sequestosome 1(SQSTM1) are widely used proteins to evaluate autophagy in tumor tissues. However, the role of MAP1LC3B and SQSTM1 in OSCC is not fully understood, particularly in certain subsites. With a tissue microarray comprised of 498 OSCC patients, including 181 BMSCC, 244 TSCC, and 73 LSCC patients, we found that the expression levels of MAP1LC3B and cytoplasmic SQSTM1 were elevated in the tumor tissues of three subsites compared with those in adjacent normal tissues. MAP1LC3B was associated with a poor prognosis only in TSCC. SQSTM1 was associated with poor differentiation in three subsites, while the association with lymph node invasion was only observed in BMSCC. Interestingly, MAP1LC3B was positively correlated with SQSTM1 in the tumor tissues of BMSCC, whereas it showed no correlation with SQSTM1 in adjacent normal tissue. The coexpression of higher MAP1LC3B and SQSTM1 demonstrated a significantly worse disease-specific survival (DSS) and disease-free survival (DFS) in patients with BMSCC and LSCC, but not TSCC. The knockdown of MAP1LC3B and SQSTM1 reduced autophagy, cell proliferation, invasion and tumorspheres of BMSCC cells. Additionally, silencing both MAP1LC3B and SQSTM1 enhanced the cytotoxic effects of paclitaxel in the tumorspheres of BMSCC cells. Taken together, MAP1LC3B and SQSTM1 might modulate autophagy to facilitate tumorigenesis and chemoresistance in OSCC, particularly in BMSCC.

## 1. Introduction

Oral squamous cell carcinoma (OSCC), a type of head and neck cancer, is one of the most common malignant tumors worldwide [1,2]. Oral cancer mainly originates from the epithelium of the oral cavity, of which the tongue, buccal mucosa, and lip are the top three most common subsites [3]. Oral cancer is a multistep process modulated by environmental and endogenous factors, such as alcohol and tobacco and betel chewing. Other factors include poor oral hygiene and chronic infections caused by viruses or bacteria. Although the standard treatment is effective for patients diagnosed at the early stage, the morbidity rate for patients with advanced-stage disease has not decreased much in the past few decades [4], requiring more precise biomarkers for either an early diagnosis or therapeutic targets for a better outcome.

Autophagy is a clearance pathway that involves more than 38 autophagy-related (ATG) proteins to recruit impaired proteins and organelles for bulk degradation for new synthesis [5]. Autophagy plays a crucial role in physiological homeostasis, and its dysfunction may cause various diseases, such as cancer, neurodegeneration disease and infection. However, the role of autophagy in tumor progression is a “double-edged sword”, with opposite functions in tumor imitation and malignancy. Autophagy exhibits suppressive effects on chronic inflammation and ROS production, thereby inhibiting carcinogenesis in the early phase [6,7] and facilitates cancer cell growth and survival under microenvironmental stress conditions [8]. Autophagy can support tumor cell survival through suppressing the p53 response and maintaining mitochondrial metabolism to prevent metabolic stress and mitigate the accumulation of toxic substances [9,10,11]. Regarding the clinical association of autophagy markers, microtubule-associated light chain 3B (MAP1LC3B), an essential protein for autophagosome elongation, is associated with poor survival in various cancer types; however, some studies have indicated that cancer patients with high MAP1LC3B expression have a better outcome [12], particularly in K-Ras-mutated colorectal cancer cells. Additionally, SQSTM1 contains an MAP1LC3B-interacting (LIR) domain and a ubiquitin-associated (UBA) domain, which serve as an autophagy adaptor to recruit ubiquitinated proteins to the autophagosome for selective autophagy [13,14]. High levels of cytoplasmic SQSTM1 have also been found to be associated with poor survival in several cancer types [12]. However, little is known about the detailed clinical relevance and functions of MAP1LC3B and SQSTM1 in certain subsites of OSCC. In this study, we compared the MAP1LC3B and SQSTM1 protein levels in tumor tissues and adjacent normal tissues in three major subsites of OSCC, including BMSCC, TSCC and LSCC. Our results show that both MAP1LC3B and cytoplasmic SQSTM1 were elevated in tumor tissues in three subsites of OSCC compared with that in adjacent normal tissues. Moreover, SQSTM1 was associated with poor survival in patients with BMSCC, but not in those with TSCC and LSCC. The SQSTM1 protein level was also found to be positively correlated with MAP1LC3B in the tumor tissues of BMSCC but not in adjacent normal tissues. In contrast to the single expression of MAP1LC3B or SQSTM1, high coexpression of MAP1LC3B and SQSTM1 showed a worse DSS and DFS in BMSCC. Silencing MAP1LC3B and SQSTM1 diminished autophagy, cell proliferation, invasion, tumorsphere formation and paclitaxel resistance in BMSCC cell lines, supporting our findings in clinical samples. Our results suggest that MAP1LC3B and SQSTM1 could serve as biomarkers or therapeutic targets for BMSCC.

## 2. Experimental Procedure

### 2.1. Tissue Specimens and Tissue Microarray (TMA) Construction

In total, 498 margin-free (margin-size ≥ 0.2 cm) paraffin-embedded materials of primary BMSCC (*n* = 181), TSCC (*n* = 244), and LSCC (*n* = 73) were established previously [3]. The data of sex, age, cell differentiation, pathological stage, tumor TNM classification, tumor subsites, and tumor recurrence time were also collected. Pathologic TNM classification was determined according to the guidelines of the 2002 American Joint Committee on Cancer (AJCC) system. The Institutional Review Board at Kaohsiung Veterans General Hospital (KVGH) approved this study to comply with the Declaration of Helsinki (IRB number: VGHKS 11-CT12-13). All information was obtained from the archives of the KVGH pathology department between 1993 and 2006.

The TMA block contained 144 cores, including 48 trios consisting of 2 cores from the tumor tissue and 1 core from the adjacent normal tissue. After construction, TMA blocks were cut in 4-μm paraffin sections using standard techniques [3].

### 2.2. Immunohistochemistry (IHC)

TMA blocks were cut into 4-μm paraffin sections for immunostaining processes as previously reported [15]. Antigen retrieval was performed by a pressure boiler at 125 °C for 10 min in Tris-EDTA (10 mM, pH 9.0) for MAP1LC3B and sodium citrate (10 mM, pH 6.0) for SQSTM1. After blocking with 3% hydrogen peroxide in methanol, the slides were incubated with antibody against MAP1LC3B (dilution 1:100; 5F10; NanoTools, Munich, Germany) and SQSTM1 (dilution 1:1000; BML-PW9860; Enzo Life Sciences, Farmingdale, NY, USA) in a cold room overnight. The color was developed at room temperature, and the sections were counterstained with hematoxylin.

### 2.3. Immunohistochemistry Analysis and Score

All the slides were independently reviewed by an oral cancer pathologist and a senior pathology technician. Subsequently, 5%–20% of core samples were randomly selected for re-evaluation. If disagreement occurred (intensity score discrepancy >1 or percentage level >20%), the slide was re-evaluated to obtain a consensus diagnosis by a senior pathologist until all the discrepancies was resolved. During the evaluation, none of them were aware of the clinical outcomes of the patients. The scores for cytoplasmic staining were based on the staining intensity (0, no signal; 1, mild; 2, moderate; and 3, strong) and percentage of positive staining (0, <5%; 1, 5%–25%; 2, 26%–50%; 3, 51%–75%; and 4, >75%). In our preliminary test, the intensity score of staining for LC3 and cytoplasmic SQSTM1 was measured and standardized (0, no expression; 1, weak expression; 2, moderate expression; and 3, strong expression; Figure 1) in OSCC. The final score, ranging from 0 to 7, was used to analyze the association of MAP1LC3B and SQSTM1 with clinicopathological features. For survival analysis, the expression levels were dichotomized as low expression and high expression with the cutoff based on the receiver operating characteristic (ROC) curve. The cutoff values were determined for MAP1LC3B and SQSTM1 in BMSCC, TSCC, LSCC and OSCC.

### 2.4. Cell Culture and Transient Transfection

The buccal mucosal squamous cell lines TW2.6 and OC3 or OC3-I5 (gift of Dr. Lu-Hai Wang) [16] were cultured in DMEM/F12 (Gibco, Life Technologies, CA, USA) with 10% FBS, 100 μg/mL of streptomycin, 100 U/mL of penicillin, and 1% L-glutamine at 37 °C with 5% CO_2_:95% air. The cells were cultured in Corning tissue culture-treated plastic dishes (Corning, Inc., Corning, NY, USA). BMSCC cells were seeded with RNAiMAX (13778150; Life Technologies, CA, USA) in the presence of 5 nM scrambled siRNA or siRNA against human MAP1LC3B (L-012846-00-0005; Dharmacon, IL, USA) or SQSTM1 (L-010230-00-0005; Dharmacon, IL, USA) for 48 h. The knockdown efficiency was determined with immunoblotting using anti-MAP1LC3B or SQSTM1 antibody as previously reported [17].

### 2.5. Analysis of Cell Viability

The cell viability assay was performed using the CellTiter-Glo luminescent cell viability assay kit (Promega, Madison, WI, USA). Briefly, 5–7 × 10^5^ cells/mL were cultured in sterile 96-well plates for 24–48 h, and then 100 μL of CellTiter-Glo reagent was added to lyse the cells for 10 min. The luminescence signal was measured using a Fluoroskan Ascent FL reader (Thermo Fisher Scientific, Waltham, CA, USA). For tumorsphere formation, the cells (OC3: 5 × 10^3^ cells/well, TW2.6: 5 × 10^3^ cells/well) were seeded into Nano Culture Plates (NCPs) (MBL Corporation, Nagoya, Japan) in the presence of siRNA for 7 days to form spheroid cells that might induce stem cell-like properties. The tumorspheres were then treated with or without the anticancer drug paclitaxel (Selleckchem, Houston, TX, USA) for two days. The sphere viability was measured using the CellTiter Glo 3D system (Promega, Madison, WI, USA).

### 2.6. Invasion Assay

For the wound-healing assay, Transwell invasion assays were performed using 8-μm pore inserts (Greiner Bio-One, Stroud, UK). The cells were knocked down with siRNA for 48 h and then were seeded into the top chamber of Transwell plates coated with 0.5% Matrigel in 300 μL of DMEM containing 1% FBS. To the bottom wells were added with complete medium to stimulate invasion. After seeding for 24 h, the cells were fixed and stained with 0.1% crystal violet. The cells that had invaded through the Matrigel and had reached to the reverse side were pictured under a microscope at a magnification of 200× and were quantified with ImageJ. Each assay was performed in triplicate.

### 2.7. Statistical Analysis

The Kruskal–Wallis one-way ANOVA test was used to evaluate the differential protein expression in TAN tissues and tumor tissues in various OSCC types. The protein expression levels between TAN tissues and tumor tissues were analyzed by the Wilcoxon signed-rank test. Student’s t test, Mann–Whitney U test, Kruskal–Wallis one-way ANOVA test and one-way ANOVA test were used to evaluate the correlation between each protein expression level and clinicopathologic parameters. The Kaplan–Meier method was used to analyze cumulative survival curves, and survival curve analysis was performed using the log-rank test. The Cox proportional hazards model was used to evaluate the impact of the protein expression on survival using factors significant in univariate analysis as covariates. The association between cell differentiation and the relative protein expression levels in tumor tissues compared with that in paired TAN tissues of individual OSCC patients with different AJCC pathological stages was determined by Fisher’s exact test. A two-sided value of *p* < 0.05 was considered statistically significant.

## 3. Results

### 3.1. Association of the MAP1LC3B and SQSTM1 Protein Levels with Tumorigenesis and Clinicopathological Outcomes

MAP1LC3B can be divided into two major forms: cytosolic MAP1LC3B-I and autophagosome membrane-bound MAP1LC3B-II (MAP1LC3B puncta), which indicates autophagy. We initially checked for MAP1LC3B puncta with IHC staining in tissues. Representative photomicrographs of MAP1LC3B and SQSTM1 for negative (0), weak (1+), moderate (2+), and strong (3+) expression in tumor tissue are shown in Figure 1. However, we found that only a few slides contained MAP1LC3B puncta in most of the OSCC TMA. Thus, we scored the total MAP1LC3B expression levels in all tissues and found that the MAP1LC3B levels were increased in tumor tissues compared with those in adjacent normal tissues in three major subsites of OSCC, including BMSCC (3.31 ± 1.44 vs. 2.30 ± 1.06, *p* < 0.001), TSCC (1.48 ± 1.02 vs. 0.59 ± 0.94, *p* < 0.001) and LSCC (3.14 ± 1.43 vs. 2.32 ± 0.71, *p* < 0.001) (Table 1). Although SQSTM1 was found in both the nucleus and cytoplasm of cells, SQSTM1 interacts with MAP1LC3B to recruit damaged proteins to the autophagosome for degradation in the lysosome via selective autophagy, suggesting SQSTM1 functions as an autophagy adaptor in nonnuclear regions. Herein, we scored the cytoplasmic SQSTM1 level of tissues to analyze its correlation with tumorigenesis and clinicopathological outcomes at three subsites of OSCC. Similar to the results of MAP1LC3B, cytoplasmic SQSTM1 was elevated in three subsites of OSCC (BMSCC: 2.89 ± 1.11 vs. 1.89 ± 1.00, *p* < 0.001; TSCC: 2.78 ± 1.08 vs. 1.88 ± 0.77, *p* < 0.001; and LSCC: 3.22 ± 1.25 vs. 2.03 ± 0.64, *p* < 0.001) (Table 1). Regarding the expression levels of MAP1LC3B and cytoplasmic SQSTM1 and clinicopathological outcomes, including cell differentiation, pathological stage, sex and age, the MAP1LC3B expression levels in BMSCC and LSCC were significantly higher than those in TSCC (*p* < 0.001, Table 2). A high MAP1LC3B protein level was associated with poor differentiation in only TSCC (*p* = 0.042, Table 2) but not in BMSCC and LSCC. On the other hand, SQSTM1 protein levels were associated with poor differentiation in BMSCC (*p* = 0.015), TSCC (*p* = 0.042) and LSCC (*p* = 0.003, Table 3). High SQSTM1 expression was correlated with lymph node invasion in BMSCC (3.22 ± 1.29 vs. 2.74 ± 1.02, *p* = 0.033, Table 3) but not in TSCC and LSCC.

### 3.2. Expression Levels of MAP1LC3B and SQSTM1 and Disease-Specific Survival (DSS) of OSCC Patients

We further determined whether MAP1LC3B expression was correlated with SQSTM1 in tumor tissues and adjacent normal tissues in three subsites of OSCC. Interestingly, Pearson’s correlation analysis showed that MAP1LC3B was positively correlated with SQSTM1 in the tumor tissues of BMSCC but showed no correlation in adjacent normal tissues (Table 4). Nevertheless, opposite effects were observed in TSCC, whereas MAP1LC3B was correlated with SQSTM1 expression in both tumor and adjacent normal tissues in LSCC (Table 4), suggesting these molecules might be differentially regulated at diverse subsites of OSCC. Moreover, to determine whether MAP1LC3B and SQSTM1 could be used as prognostic factors in the different subsites of OSCC, we further investigated the relationship of MAP1LC3B and SQSTM1 expression with DSS. Kaplan–Meier curve analysis showed that higher MAP1LC3B (Figure 2) expression was associated with a poor DSS in LSCC (*p* = 0.008), whereas high SQSTM1 (Figure 3) expression was associated with a poor DSS in BMSCC (*p* = 0.005). After adjustment for cell differentiation (moderate + poor vs. well) and AJCC pathological stage (stage III + IV vs. stage I + II) following Cox’s regression analysis, both MAP1LC3B (AHR: 1.59, 95% CI: 1.00–2.52, *p* = 0.05, Table 5) and SQSTM1 (AHR: 1.92, 95% CI: 1.19–3.09, *p* = 0.008, Table 4) were correlated with unfavorable DSS in BMSCC. High MAP1LC3B expression showed a worse DSS in LSCC (AHR: 19.93, 95% CI: 1.61–246.87, *p* = 0.02, Table 4). Additionally, BMSCC and LSCC with high coexpression of MAP1LC3B and SQSTM1 had a shorter DSS than those with low coexpression of MAP1LC3B and SQSTM1 (BMSCC: *p* = 0.019, LSCC: *p* = 0.012, Figure 3). After adjusting for cell differentiation and AJCC pathological stage, the results also showed that BMSCC (AHR: 2.38, 95% CI: 1.27–4.46, *p* = 0.007, Table 5) and LSCC (AHR: 20.72, 95% CI: 1.72–250.12, *p* = 0.017, Table 5) patients with high coexpression of MAP1LC3B and SQSTM1 had higher hazard of death.

### 3.3. Association of the MAP1LC3B and SQSTM1 Expression Levels with Disease-Free Survival (DFS) in OSCC Patients.

To determine whether MAP1LC3B and SQSTM1 are correlated with relapse in the main subsites of OSCC, we further investigated the relationship of MAP1LC3B and SQSTM1 expression with DFS. The Kaplan–Meier curve showed that a high expression level of MAP1LC3B was notably associated with a poor DFS in BMSCC (*p* = 0.023, Figure 4) and LSCC (*p* = 0.009, Figure 4). High expression of SQSTM1 was associated with shorter DFS in only LSCC (*p* = 0.007) but not at the other two subsites. Likewise, the hazard factor was higher in patients with an elevated expression of MAP1LC3B after adjustment for cell differentiation and AJCC pathological stage following Cox’s regression analysis in BMSCC (AHR: 1.90, 95% CI: 0.94–3.85, *p* = 0.074, Table 6). High SQSTM1 expression had a poor DFS in BMSCC (AHR: 3.77, 95% CI: 1.06–13.38, *p* = 0.040, Table 6). The combination of high MAP1LC3B and SQSTM1 expression was highly associated with a shorter DFS in patients with BMSCC (*p* < 0.001, Figure 5) and LSCC (*p* = 0.003, Figure 5). The adjusted hazard ratios of high coexpression of MAP1LC3B and SQSTM1 in BMSCC and LSCC were also much higher than those with a high single expression of MAP1LC3B or SQSTM1(BMSCC: AHR: 8.19, 95% CI: 2.52–26.64, *p* < 0.001; LSCC: AHR: 9.49, 95% CI: 2.19–41.08, *p* = 0.003, Table 6).

### 3.4. Involvement of MAP1LC3B and SQSTM1 in the Cell Proliferation and Migration of BMSCC Cells.

According to the clinical results described above, MAP1LC3B and SQSTM1 were more correlated with cancer malignancy in BMSCC and LSCC. Because there was no LSCC cell line available, we further verified the function of MAP1LC3B and SQSTM1 in BMSCC cell lines, including OC3 and TW2.6 (Figure 6A,B). Knockdown of MAP1LC3B resulted in accumulated SQSTM1 in BMSCC cells, while silencing SQSTM1 decreased MAP1LC3B-II flux, implying that deprivation of either MAP1LC3B and SQSTM1 diminished autophagic flux (Figure 6C,D). Similar to treatment with the autophagy inducer ConA, knockdown of MAP1LC3B or SQSTM1 attenuated cell proliferation in both OC3 and TW2.6 cells (Figure 7A). Autophagy is involved in cancer metastasis, and SQSTM1 is associated with lymph node invasion. To determine the effects of MAP1LC3B and SQSTM1 on metastatic characteristics, invasion assays were used. Silencing SQSTM1 inhibited both the invasive ability in highly invasive OC3 cells (OC3-I5, Figure 7B, Wang LH 2017, ROS1). We further mimicked the in vivo status in a tumorsphere culture model for cancer cell stemness (REF) and drug resistance (Figure 7C,D). Interestingly, ablation of MAP1LC3B or SQSTM1 or both decreased tumorsphere formation and enhanced the killing effects of paclitaxel (PTX, Figure 7C,D), implying that MAP1LC3B and SQSTM1 might modulate autophagy for tumor growth and drug resistance in BMSCC cells.

## 4. Discussion

MAP1LC3B and SQSTM1 are widely used autophagy markers and adaptors in mammalian cells. Both proteins are essential for the autophagy machinery, and high expression levels of MAP1LC3B and SQSTM1 are significantly associated with an unfavorable clinicopathological outcome in several cancer types, including OSCC. However, the subsite-dependent impact of MAP1LC3B and SQSTM1 on OSCC, such as BMSCC, TSCC and LSCC, is not fully understood. Moreover, the function of MAP1LC3B and SQSTM1 in OSCC remains unclear. In this study, we reported the following findings. First, the expression levels of MAP1LC3B and SQSTM1 were higher in tumor tissues than in adjacent normal tissues at three subsites—BMSCC, TSCC and LSCC. Second, SQSTM1 was correlated with aggressive differentiation in three subsites and was associated with lymph node invasion in BMSCC. Third, SQSTM1 was positively correlated with MAP1LC3B in the tumor tissues of BMSCC, but not in adjacent normal tissues. High coexpression of MAP1LC3B and SQSTM1 was further associated with a poor survival, particularly in BMSCC and LSCC. Fourth, silencing of MAP1LC3B and SQSTM1 diminished autophagy, cell proliferation, and invasion and sensitized BMSCC cells to paclitaxel treatment. Our results suggested that MAP1LC3B and SQSTM1 may modulate autophagy for cancer development, malignancy and relapse in a subsite-dependent manner. To the best of our knowledge, we are the first group to report that the correlation of MAP1LC3B and SQSTM1 with clinicopathological outcomes at certain subsites using the largest cohort, comprising 498 paired tumor and adjacent normal tissues of OSCC.

MAP1LC3B is cleaved and activated by ATG4 to mediate a ubiquitination-like reaction to initiate autophagosome formation [18,19,20]. MAP1LC3B consists of a soluble form, MAP1LC3B-I, with a molecular weight of 18 KD and a membrane-bound form, MAP1LC3B-II, with a molecular weight of 16 KD. MAP1LC3B-II accumulates on the autophagosome and interacts with SQSTM1 for recruitment to damaged proteins to deliver them to lysosomes [21]. Thus, both MAP1LC3B-II and SQSTM1 can form puncta on autophagosomes during selective autophagy [21]. MAP1LC3B-II and SQSTM1 dot-like staining are shown in tissues and are associated with a poor prognosis in patients with colon cancer [22]. Increased MAP1LC3B puncta is also associated with a poor prognosis in several other cancer types, such as breast cancer and oral cancer [23,24]. However, the puncta of both MAP1LC3B-II and SQSTM1 were very rare in all tissues of our TMA, likely due to the different cancer types or tissues that we used. Moreover, increased MAP1LC3B expression typically shows unfavorable outcomes in lung, melanoma, and pancreatic cancers [25,26], whereas the loss of MAP1LC3B has been reported in several solid tumors, including brain cancer [27], prostate cancer [28], and breast cancer, indicating that MAP1LC3B expression in cancer is still controversial. In our present study, MAP1LC3B expression was higher in tumor tissues than in adjacent normal tissues at three subsites of OSCC. MAP1LC3B was positively correlated with DSS in patients with BMSCC and LSCC, but not in those with TSCC and OSCC. Deprivation of MAP1LC3B with siRNA suppressed cell viability and tumorsphere formation in BMSCC cell lines, implying that MAP1LC3B may contribute to tumorigenesis and malignancy at certain subsites of OSCC, particularly in BMSCC.

Autophagy induction causes SQSTM1 degradation, while defective autophagy leads to SQSTM1 accumulation. Higher SQSTM1 expression is associated with a poor prognosis in gastric cancer [29]. Previous studies have also reported that higher expression of SQSTM1 is correlated with a worse survival in several solid tumors [30,31]. These results suggest that autophagy impairment accumulates SQSTM1, ultimately leading to tumorigenesis by dysregulating the NF-κB signaling transduction pathway and gene expression [14,32]. The interaction of SQSTM1 with tumor necrosis factor receptor-associated factor (TRAF) 6, as well as the degradation of SQSTM1 by autophagy, is important for the role of SQSTM1 in tumorigenesis and cell survival [32]. Nevertheless, the correlation of SQSTM1 with detailed subsites of cancers has never been reported. Our present data show that, at three subsites of OSCC, cytoplasmic SQSTM1 protein expression in TAN tissue was significantly lower than that in tumor tissue. Additionally, SQSTM1 was significantly associated with N classification following Student’s t test in BMSCC. Silencing SQSTM1 inhibited cancer cell invasion in BMSCC cells. In the Cox regression method, higher-level expression of cytoplasmic SQSTM1 was associated with a poor prognosis in OSCC, mainly in BMSCC and LSCC, implying that SQSTM1 could be an independent biomarker of the prognosis in BMSCC and LSCC.

The role of autophagy could be switched from a tumor suppressor to an oncogene during tumor progression. Autophagy acts as a tumor suppressor to eliminate abnormal proteins or organelles and reduce the production of reactive oxygen species and DNA damage in the early stage of cancer development [33]. Prodigiosin, a red pigment isolated from gram-negative bacteria, induces autophagic cell death in both lung and oral cancer cells [34,35]. By contrast, autophagy allows cancer cells to survive during metastasis and chemotherapy, which, in turn, results in tumor relapse [36,37]. Autophagic activity is higher in cancer stem cells (CSC) of ovarian cancer than in non-CSC [37]. Pharmacologic inhibition with the autophagy inhibitor chloroquine or genetic ablation with CRISPR/Cas9 knockout for the autophagy essential gene ATG5 significantly reduced the CSC property and chemoresistance of ovarian cancer cells. Pharmacologic inhibition of the essential autophagy protease ATG4 with clinical drug tioconazole suppresses the tumor size and sensitizes cancer cells to doxorubicin [5]. Moreover, autophagy inhibition decreases metastatic outbreak and increases apoptosis in dormant BC cells [36]. Our study shows that higher cytoplasmic SQSTM1 expression is correlated with lymph node invasion in patients with BMSCC. The silencing of SQSTM1 reduces autophagic flux and invasion in BMSCC cells. SQSTM1 expression was positively correlated with MAP1LC3B expression in tumor tissues of patients with BMSCC. Furthermore, high expression of both MAP1LC3B and SQSTM1 was associated with a shorter DSS and DFS in patients with BMSCC and LSCC, but not in those with TSCC patients. Similar to treatment with autophagy inhibitor, knockdown of these genes repressed autophagy, cell proliferation and chemoresistance in BMSCC cells. Our present results suggested that autophagy may act as a tumor promoter at certain subsites of oral cancer, particularly in BMSCC.

On the other hand, autophagy diminishes the cytotoxic effects of T cells and natural killer (NK) cells against tumor cells [38,39]. PD-L1 inhibits autophagy through activation of MTOR, while the PD-L1 inhibitor attenuates autophagy for cancer cell survival [40]. The anti-malarial drug CQ or HCQ, which blocks autophagosome-lysosome fusion and degradation, has been tested in at least 30 clinical trials for cancer [41]. These results suggest that blocking autophagy might be helpful for cancer therapy of OSCC, which requires further study to evaluate.

## Figures and Tables

**Figure 1 jcm-07-00478-f001:**
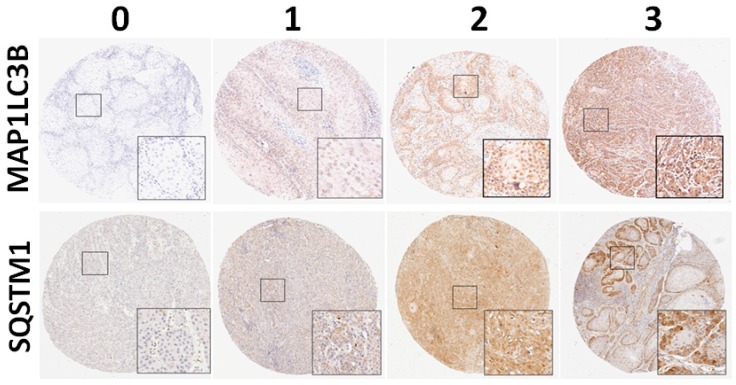
Protein levels of MAP1LC3B puncta and SQSTM1 in OSCC. The MAP1LC3B puncta and cytoplasmic SQSTM1 were stained by immunohistochemistry and categorized into four different levels as follows: 0 = negative staining; 1 = weak; 2 = moderate; 3 = strong.

**Figure 2 jcm-07-00478-f002:**
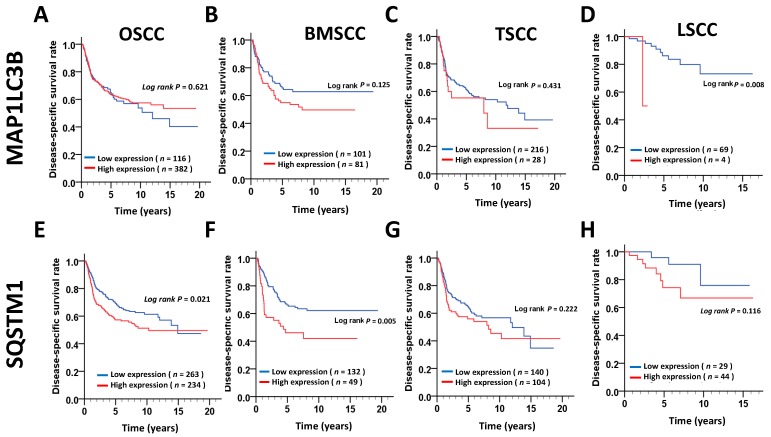
DSS survival curves for MAP1LC3B and SQSTM1 expression in patients with different subsites of OSCC. DSS survival curves of MAP1LC3 (**A**–**D**) and SQSTM1 (**E**–**H**) are shown for OSCC (**A**,**E**) and three main subsites, BMSCC (**B**,**F**), TSCC (**C**,**G**) and LSCC (**D**,**H**). The cutoff values for high or low expression of MAP1LC3B and SQSTM1 in tumor tissues were based on the receiver operating characteristic (ROC) curve.

**Figure 3 jcm-07-00478-f003:**
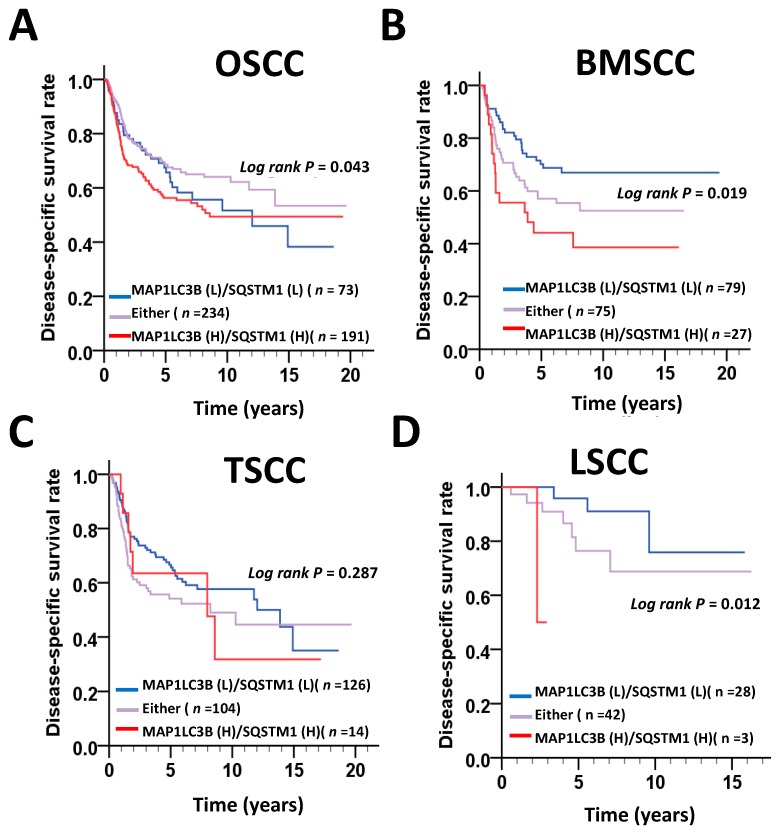
DSS survival curves for the coexpression of MAP1LC3B and SQSTM1 in patients with different subsites of OSCC. DSS survival curves for the coexpression of MAP1LC3 and SQSTM1 are shown for OSCC (**A**) and three main subsites, BMSCC (**B**), TSCC (**C**) and LSCC (**D**). The cutoff values for high or low coexpression of MAP1LC3B and SQSTM1 in tumor tissues were based on the receiver operating characteristic (ROC) curve.

**Figure 4 jcm-07-00478-f004:**
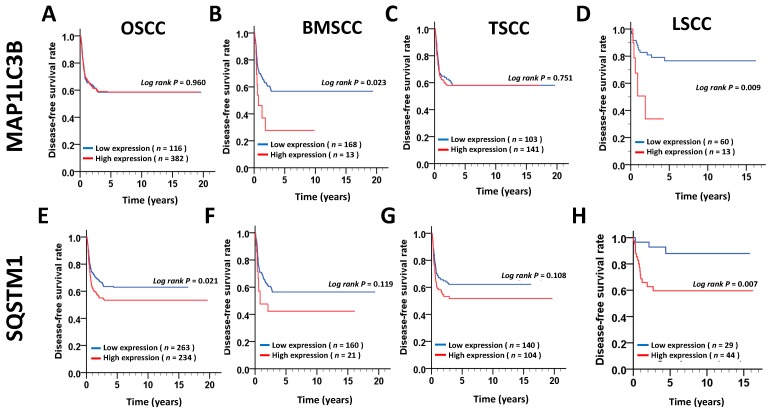
DFS survival curves for MAP1LC3B and SQSTM1 expression in patients with different subsites of OSCC. DFS survival curves of MAP1LC3 (**A**–**D**) and SQSTM1 (**E**–**H**) are shown for OSCC (**A**,**E**) and three main subsites, BMSCC (**B**,**F**), TSCC (**C**,**G**) and LSCC (**D**,**H**). The cutoff values for high or low expression of MAP1LC3B and SQSTM1 on tumor tissues were based on the receiver operating characteristic (ROC) curve.

**Figure 5 jcm-07-00478-f005:**
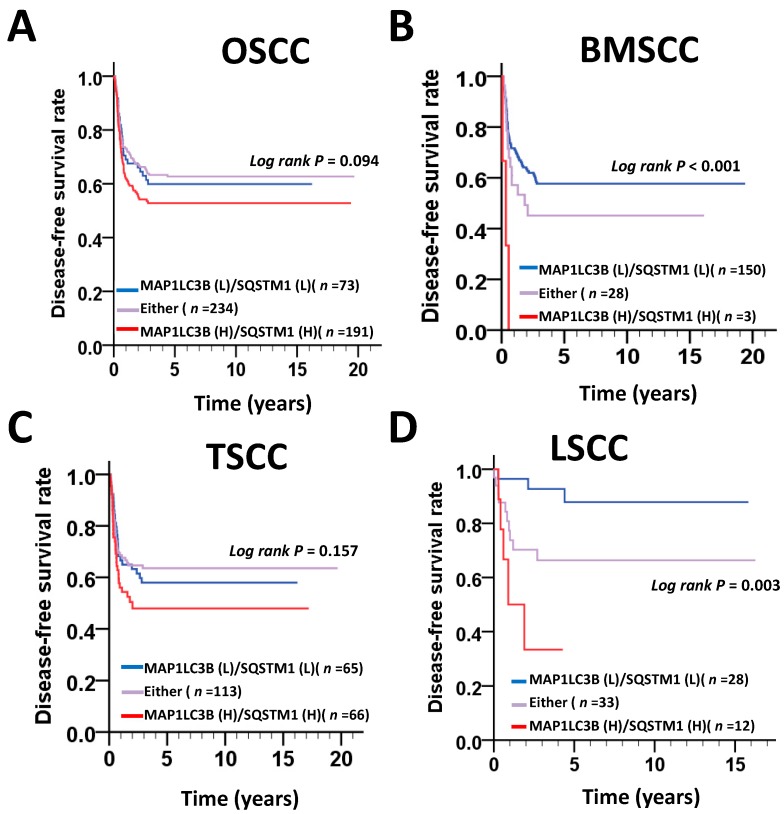
DFS survival curves for the coexpression of MAP1LC3B and SQSTM1 in patients with different subsites of OSCC. DFS survival curves for the coexpression of MAP1LC3 and SQSTM1 are shown for OSCC (**A**) and three main subsites, BMSCC (**B**), TSCC (**C**) and LSCC (**D**). The cutoff values for high or low coexpression of MAP1LC3B and SQSTM1 in tumor tissues were based on the receiver operating characteristic (ROC) curve

**Figure 6 jcm-07-00478-f006:**
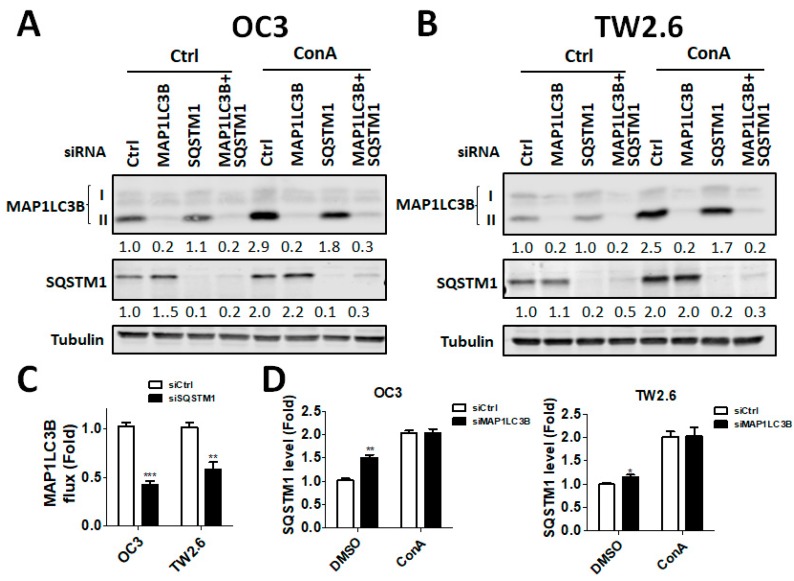
Effects of silencing MAP1LC3B and SQSTM1 on autophagy and cell proliferation in BMSCC cells. (**A**) The BMSCC cell lines OC3 and (**B**) TW2.6 were transfected with 5 nM scrambled siRNA or siRNA against MAP1LC3B or SQSTM1 for 48 h and then were treated with ConA for 4 h. The knockdown efficiency was determined by immunoblotting. (**C**) Silencing effects on MAP1LC3B flux and (**D**) SQSTM1 levels were quantified and analyzed.

**Figure 7 jcm-07-00478-f007:**
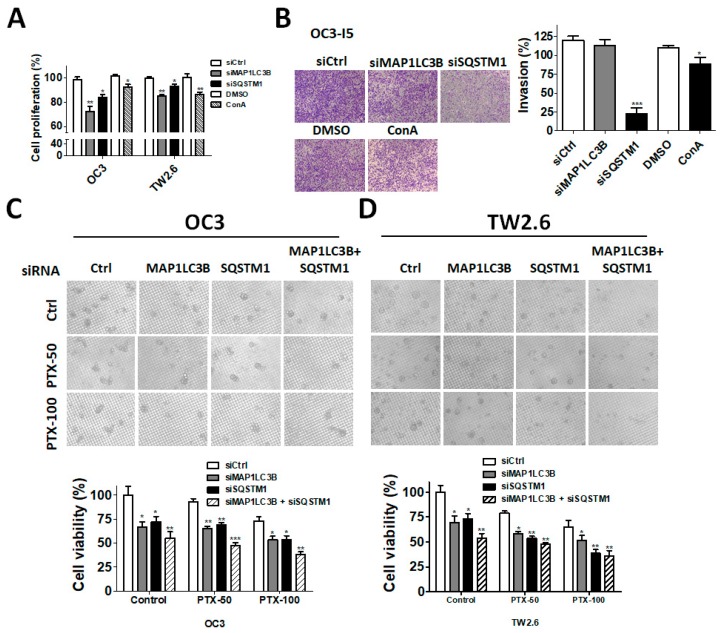
Effects of silencing MAP1LC3B and SQSTM1 on metastatic characteristics, tumorsphere formation and paclitaxel sensitivity in BMSCC cells. (**A**) The BMSCC cell lines OC3 and TW2.6 were transfected with 5 nM scrambled siRNA or siRNA against MAP1LC3B or SQSTM1 for 72 h to measure cell proliferation with Cell titer Glo. The cells treated with the autophagy inhibitor ConA for 24 h were used as a control. (**B**) OC3-I5 cells, the highly invasive strain of OC3, were silenced for 48 h and were seeded in Matrigel-coated Transwell filters to assess the cell invasion of BMSCC cells. (**C**) The silenced OC3 and (**D**) TW2.6 cells were cultured in nanoplates to examine tumorsphere formation. The tumorspheres were also treated with or without paclitaxel (PTX, 50 or 100 nM) to determine the effects of genes on drug resistance. The results represented three independent experiments.

**Table 1 jcm-07-00478-t001:** Comparisons of MAP1LC3B and SQSTM1 expression between tumor tissues and corresponding tumor adjacent normal tissues at three subsites of oral SCC.

Variables	No.	Tumor adjacent normal		Tumor		Z	*p*-value *
		Mean ± SD	Median	Mean ± SD	Median		
BMSCC							
MAP1LC3B	135	2.01 ± 1.08	2.00	3.31 ± 1.44	3.00	7.787	**<0.001**
SQSTM1	132	1.89 ± 1.00	2.00	2.89 ± 1.11	2.00	6.821	**<0.001**
TSCC							
MAP1LC3B	174	0.59 ± 0.94	0.00	1.48 ± 1.02	2.00	7.710	**<0.001**
SQSTM1	192	1.88 ± 0.77	2.00	2.78 ± 1.08	2.00	7.939	**<0.001**
LSCC							
MAP1LC3B	59	2.32 ± 0.71	2.00	3.14 ± 1.43	3.00	4.084	**<0.001**
SQSTM1	59	2.03 ± 0.64	2.00	3.22 ± 1.25	3.00	5.049	**<0.001**
OSCC							
MAP1LC3B	368	1.39 ± 1.23	2.00	2.42 ± 1.54	2.00	11.717	**<0.001**
SQSTM1	383	1.91 ± 0.84	2.00	2.89 ± 1.13	2.00	11.620	**<0.001**

Abbreviations: SCC, squamous cell carcinoma; SD, standard deviation. * *p*-values were estimated by the Wilcoxon signed-rank test. Bold values denote statistically significant.

**Table 2 jcm-07-00478-t002:** Expression of MAP1LC3B and clinicopathologic outcomes in patients with OSCC and three primary subsites.

Variable	BMSCC (*n* = 181)				TSCC (*n* = 244)				LSCC (*n* = 73)				OSCC (*n* = 498)			
	%	Mean ± SD	Median	*p*-value	%	Mean ± SD	Median	*p*-value	%	Mean ± SD	Median	*p*-value	%	Mean ± SD	Median	*p*-value
Sex																
Female	2.2	2.50 ± 0.58	2.50	0.240 *	11.9	1.69 ± 1.34	2.00	0.311 *	9.6	3.71 ± 0.95	4.00	0.246 *	8.0	2.13 ± 1.44	2.00	0.256 *
Male	97.8	3.34 ± 1.42	3.00		88.1	1.47 ± 1.03	2.00		90.4	3.00 ± 1.58	3.00		92.0	2.41 ± 1.56	2.00	
Age, year																
≤50	44.2	3.39 ± 1.31	3.00	0.570 *	51.6	1.52 ± 1.04	2.00	0.812 *	21.9	2.69 ± 1.62	2.00	0.266 *	44.6	2.27 ± 1.48	2.00	0.131 *
>50	55.8	3.27 ± 1.49	3.00		48.4	1.48 ± 1.11	2.00		78.1	3.18 ± 1.51	3.00		55.4	2.49 ± 1.60	2.00	
Subsite																
Buccal	100.0	3.32 ± 1.41	3.00	-	-	-	-	-	-	-	-	-	36.3	3.32 ± 1.41 ^a^	3.00	**<0.001 ^†^**
Tongue	-	-	-		100.0	1.50 ± 1.07	2.00		-	-	-		49.0	1.50 ± 1.07 ^ab^	2.00	
Lip	-	-	-		-	-	-		100.0	3.07 ± 1.54	3.00		14.7	3.07 ± 1.54 ^b^	3.00	
Cell differentiation																
Well	26.0	3.04 ± 1.38	3.00	0.213 ^‡^	10.7	1.08 ± 1.06 ^c^	1.00	**0.042 ^‡^**	46.6	3.00 ± 0.98	3.00	0.726 ^†^	21.5	2.55 ± 1.45	2.00	0.209 ^‡^
Moderate	69.1	3.39 ± 1.40	3.00		82.4	1.52 ± 1.06	2.00		47.9	3.06 ± 1.80	3.00		72.5	2.32 ± 1.55	2.00	
Poor	5.0	3.78 ± 1.56	3.00		7.0	1.88 ± 1.05 ^c^	2.00		5.5	3.75 ± 2.99	4.00		6.0	2.70 ± 1.76	2.00	
AJCC pathological stage																
I, II	61.3	3.30 ± 1.33	3.00	0.782 *	68.9	1.45 ± 1.05	2.00	0.246 *	79.5	3.21 ± 1.48	3.00	0.132 *	67.7	2.36 ± 1.53	2.00	0.498 *
III, IV	38.7	3.36 ± 1.53	3.00		31.1	1.62 ± 1.12	2.00		20.5	2.53 ± 1.68	2.00		32.3	2.46 ± 1.59	2.00	
T classification																
T1, T2	75.7	3.25 ± 1.38	3.00	0.224 *	79.5	1.47 ± 1.04	2.00	0.460 *	82.2	3.17 ± 1.47	3.00	0.245 *	78.5	2.36 ± 1.52	2.00	0.321 *
T3, T4	24.3	3.55 ± 1.50	4.00		20.5	1.60 ± 1.18	2.00		17.8	2.62 ± 1.80	3.00		21.5	2.52 ± 1.66	2.00	
N classification																
N0	75.1	3.26 ± 1.36	3.00	0.296 *	79.9	1.45 ± 1.07	2.00	0.157 *	94.5	3.12 ± 1.57	3.00	0.277 *	80.3	2.35 ± 1.54	2.00	0.256 *
N1, N2	24.9	3.51 ± 1.55	3.00		20.1	1.69 ± 1.08	2.00		5.5	2.25 ± 0.50	2.00		19.7	2.55 ± 1.57	2.00	

Abbreviations: SCC, squamous cell carcinoma; AJCC, American Joint Committee on Cancer. * *p* values were estimated by student’s t-test. ^†^
*p* values were estimated by Kruskal–Wallis one-way ANOVA test. ^‡^
*p* values were estimated by one-way ANOVA test. ^a^
*p* < 0.001; ^b^
*p* < 0.001; ^c^
*p* = 0.054. Bold values denote statistically significant.

**Table 3 jcm-07-00478-t003:** Expression of SQSTM1 and clinicopathologic outcomes in patients with OSCC and three primary subsites.

Variable	BMSCC (*n* = 181)				TSCC (*n* = 244)				LSCC (*n* = 73)				OSCC (*n* = 498)			
	%	Mean ± SD	Median	*p*-value	%	Mean ± SD	Median	*p*-value	%	Mean ± SD	Median	*p*-value	%	Mean ± SD	Median	*p*-value
Sex																
Female	2.2	3.50 ± 1.73	3.50	0.246 *	11.9	3.03 ± 1.27	3.00	0.155 *	9.6	3.29 ± 1.11	3.00	0.622 *	8.0	3.13 ± 1.26	3.00	0.099 *
Male	97.8	2.85 ± 1.09	2.00		88.1	2.73 ± 1.05	2.00		90.4	3.03 ± 1.31	3.00		92.0	2.82 ± 1.11	2.00	
Age, year																
≦50	44.2	2.80 ± 1.10	2.00	0.506 *	51.6	2.68 ± 1.06	2.00	0.211 *	21.9	3.00 ± 1.10	3.00	0.849 *	44.6	2.75 ± 1.07	2.00	0.089 *
>50	55.8	2.91 ± 1.12	3.00		48.4	2.86 ± 1.10	2.00		78.1	3.07 ± 1.35	3.00		55.4	2.92 ± 1.16	3.00	
Subsite																
Buccal	100.0	2.86 ± 1.11	2.00	-	-	-	-	-	-	-	-	-	36.3	2.86 ± 1.11	2.00	0.152 ^†^
Tongue	-	-	-		100.0	2.77 ± 1.08	2.00		-	-	-		49.0	2.77 ± 1.08	2.00	
Lip	-	-	-		-	-	-		100	3.05 ± 1.29	3.00		14.7	3.05 ± 1.29	3.00	
Cell differentiation																
Well	26.0	2.53 ± 0.97 ^a^	2.00	**0.015 ^†^**	10.7	2.58 ± 0.95 ^b^	2.00	**0.040 ^‡^**	46.6	2.53 ± 1.02 ^d^	2.00	**0.003 ^†^**	21.5	2.54 ± 0.97 ^ef^	2.00	**0.001 ^‡^**
Moderate	69.1	2.94 ± 1.11	3.00		82.4	2.72 ± 1.02 ^c^	2.00		47.9	3.46 ± 1.24 ^d^	3.00		72.5	2.87 ± 1.09 ^eg^	2.00	
Poor	5.0	3.56 ± 1.33 ^a^	4.00		7.0	3.59 ± 1.58 ^bc^	4.00		5.5	4.00 ± 2.16	3.50		6.0	3.63 ± 1.54 ^fg^	4.00	
AJCC pathological stage																
I, II	61.3	2.79 ± 1.05	2.00	0.293 *	68.9	2.74 ± 1.01	2.00	0.935 ^§^	79.5	2.97 ± 1.34	3.00	0.248 *	67.7	2.80 ± 1.09	2.00	0.415 ^§^
III, IV	38.7	2.97 ± 1.19	2.00		31.1	2.82 ± 1.23	2.00		20.5	3.40 ± 1.06	4.00		32.3	2.94 ± 1.20	2.00	
T classification																
T1, T2	75.7	2.88 ± 1.11	2.00	0.765 *	79.5	2.80 ± 1.07	2.00	0.355 *	82.2	2.97 ± 1.33	3.00	0.212 *	78.5	2.85 ± 1.13	2.00	0.754 *
T3, T4	24.3	2.82 ± 1.11	2.00		20.5	2.64 ± 1.12	2.00		17.8	3.46 ± 1.05	4.00		21.5	2.81 ± 1.13	2.00	
N classification																
N0	75.1	2.74 ± 1.02	2.00	**0.033 ^§^**	79.9	2.75 ± 1.04	2.00	0.718 *	94.5	3.07 ± 1.31	3.00	0.630 *	80.3	2.81 ± 1.09	2.00	0.285 ^§^
N1, N2	24.9	3.22 ± 1.29	3.00		20.1	2.82 ± 1.24	2.00		5.5	2.75 ± 0.96	2.50		19.7	3.00 ± 1.26	2.00	

Abbreviations: SCC, squamous cell carcinoma; AJCC, American Joint Committee on Cancer. * *p* values were estimated by student’s t-test. ^†^
*p* values were estimated by one-way ANOVA test. ^‡^
*p* values were estimated by Kruskal–Wallis one-way ANOVA test. ^§^
*p* values was estimated by Mann–Whitney U test. ^a^
*p* = 0.038; ^b^
*p* = 0.031; ^c^
*p* = 0.016; ^d^
*p* = 0.008; ^e^
*p* = 0.019; ^f^
*p* < 0.001; ^g^
*p* = 0.005. Bold values denote statistically significant.

**Table 4 jcm-07-00478-t004:** Correlation coefficients (*r*) between MAP1LC3B and SQSTM1 in OSCC.

	Adjacent Normal	Tumor
	MAP1LC3B	MAP1LC3B
BMSCC	(*n* = 114)	(*n* = 181)
SQSTM1	*r* = −0.053	*r* = 0.155
	*p* = 0.578	*p* = 0.038
TSCC	(*n* = 166)	(*n* = 244)
SQSTM1	*r* = 0.162	*r* = 0.095
	*p* = 0.037	*p* = 0.140
LSCC	(*n* = 55)	(*n* = 73)
SQSTM1	*r* = 0.266	*r* = 0.504
	*p* = 0.049	*p* < 0.001
OSCC	(*n* = 365)	(*n* = 498)
SQSTM1	0.068	0.181
	*p* = 0.218	*p* < 0.001

The correlation coefficient and *p*-value were estimated by the Spearman’s rank correlation coefficient.

**Table 5 jcm-07-00478-t005:** The expression levels of MAP1LC3B and SQSTM1 in disease-specific survival of oral SCC patients.

Variable (ROC)		No. (%)	CHR (95% CI)	*p*-value	AHR (95% CI)	*p*-value *
OSCC						
MAP1LC3B expression	Low (0–1)	116 (23.3)	1.00		1.00	
	High (2–7)	382 (76.7)	0.92 (0.67–1.27)	0.621	0.99 (0.71–1.36)	0.928
SQSTM1 expression	Low (0–2)	264 (53.0)	1.00		1.00	
	High (3–7)	234 (47.0)	1.39 (1.05–1.84)	0.022	1.46 (1.10–1.94)	0.009
MAP1LC3B (L) SQSTM1 (L)		73 (14.7)	1.00		1.00	
either		234 (47.0	0.71 (0.53–0.94)	0.017	0.78 (0.52–1.18)	0.235
MAP1LC3B (H) SQSTM1 (H)		191(38.4)	1.38 (1.04–1.84)	0.026	1.15 (0.76–1.73)	0.501
BMSCC						
MAP1LC3B expression	Low (0–3)	101 (55.8)	1.00		1.00	
	High (4–7)	80 (44.2)	1.42 (0.91–2.24)	0.127	1.59 (1.00–2.52)	0.050
SQSTM1 expression	Low (0–3)	132 (72.9)	1.00		1.00	
	High (4–7)	49 (27.1)	1.96 (1.22–3.14)	0.005	1.92 (1.19–3.09)	0.008
MAP1LC3B (L) SQSTM1 (L)		79 (43.6)	1.00		1.00	
either		75 (41.4)	1.23 (0.78–1.94)	0.367	1.59 (0.95–2.67)	0.077
MAP1LC3B (H) SQSTM1 (H)		27 (14.9)	1.87 (1.07–3.24)	0.027	2.38 (1.27–4.46)	0.007
TSCC						
MAP1LC3B expression	Low (0–2)	216 (88.5)	1.00		1.00	
	High (3–7)	28 (11.5)	1.25 (0.71–2.20)	0.432	1.09 (0.62–1.91)	0.777
SQSTM1 expression	Low (0–2)	140 (57.4)	1.00		1.00	
	High (3–7)	104 (42.6)	1.27 (0.87–1.85)	0.223	1.36 (0.93–1.98)	0.117
MAP1LC3B (L) SQSTM1 (L)		126 (51.6)	1.00		1.00	
either		104 (42.6)	1.33 (0.91–1.94)	0.141	1.36 (0.92–2.01)	0.120
MAP1LC3B (H) SQSTM1 (H)		14 (5.7)	1.11 (0.51–2.38)	0.795	1.27 (0.58–2.80)	0.553
LSCC						
MAP1LC3B expression	Low (0–5)	69 (94.5)	1.00		1.00	
	High (6–7)	4 (5.5)	11.40 (1.17–111.09)	0.036	19.93 (1.61–246.87)	0.020
SQSTM1 expression	Low (0–2)	29 (39.7)	1.00		1.00	
	High (3–7)	44 (60.3)	2.79 (0.74–10.56)	0.132	2.61 (0.68–10.01)	0.161
MAP1LC3B (L) SQSTM1 (L)		28 (38.4)	1.00		1.00	
either		42 (57.5)	1.90 (0.55–6.50)	0.309	2.53 (0.65–9.82)	0.180
MAP1LC3B (H) SQSTM1 (H)		3 (4.1)	11.40 (1.17–111.09)	0.036	20.72 (1.72–250.12)	0.017

Abbreviations: SCC, squamous cell carcinoma; CHR, crude hazard ratio; CI, confidence interval; AHR, adjusted hazard ratio. * *p*-value were adjusted for cell differentiation (moderate + poor vs. well) and AJCC pathological stage (stage III + IV vs. stage I + II) by multiple Cox’s regression. Bold values denote statistically significant.

**Table 6 jcm-07-00478-t006:** The expression levels of MAP1LC3B and SQSTM1 in disease-free survival of oral SCC patients.

Variable (ROC)		No. (%)	CHR (95% CI)	*p*-value	AHR (95% CI)	*p*-value *
OSCC						
MAP1LC3B expression	Low (0–1)	116 (23.3)	1.00		1.00	
	High (2–7)	382 (76.7)	1.01 (0.73–1.40)	0.960	1.09 (0.78–1.52)	0.620
SQSTM1 expression	Low (0–2)	264 (53.0)	1.00		1.00	
	High (3–7)	234 (47.0)	1.39 (1.05–1.84)	**0.022**	1.33 (1.00–1.76)	**0.048**
MAP1LC3B (L) SQSTM1 (L)		73 (14.7)	1		1	
either		234 (47.0	0.77 (0.58–1.03)	0.075	0.93 (0.61–1.43)	0.737
MAP1LC3B(H) SQSTM1 (H)		191(38.4)	1.36 (1.03–1.81)	**0.032**	1.29 (0.84–1.98)	0.244
BMSCC						
MAP1LC3B expression	Low (0–5)	168 (92.8)	1.00		1.00	
	High (6–7)	13 (7.2)	2.20 (1.10–4.40)	**0.027**	1.90 (0.94–3.85)	0.074
SQSTM1 expression	Low (0–4)	160 (88.4)	1.00		1.00	
	High (5–7)	21 (11.6)	1.62 (0.88–3.00)	0.123	1.47 (0.79–2.72)	0.227
MAP1LC3B (L) SQSTM1 (L)		150 (82.9)	1		1	
either		28 (15.5)	1.39 (0.79–2.44)	0.251	1.45 (0.83–2.56)	0.195
MAP1LC3B(H) SQSTM1 (H)		3 (1.7)	7.66 (2.37–24.75)	**0.001**	8.19 (2.52–26.64)	**<0.001**
TSCC						
MAP1LC3B expression	Low (0–1)	103 (42.2)	1.00		1.00	
	High (2–7)	141 (57.8)	1.07 (0.71–1.59)	0.751	1.00 (0.67–1.50)	1.000
SQSTM1 expression	Low (0–2)	140 (57.4)	1.00		1.00	
	High (3–7)	104 (42.6)	1.38 (0.93–2.06)	0.109	1.35 (0.91–2.01)	0.137
MAP1LC3B (L) SQSTM1 (L)		65 (26.6)	1		1	
either		113 (46.3)	0.76 (0.51–1.14)	0.179	0.90 (0.55–1.48)	0.680
MAP1LC3B(H) SQSTM1 (H)		66 (27.0)	1.50 (0.98–2.29)	0.061	1.41 (0.84–2.36)	0.199
LSCC						
MAP1LC3B expression	Low (0–4)	60 (82.2)	1.00		1.00	
	High (5–7)	13 (17.8)	3.82 (1.31–11.13)	**0.014**	2.44 (0.78–7.65)	0.127
SQSTM1 expression	Low (0–2)	29 (39.7)	1.00		1.00	
	High (3–7)	44 (60.3)	4.75 (1.37–16.49)	**0.014**	3.77 (1.06–13.38)	**0.040**
MAP1LC3B (L) SQSTM1 (L)		28 (38.4)	1		1	
either		33 (45.2)	1.66 (0.65–4.21)	0.287	3.80 (1.04–13.86)	**0.043**
MAP1LC3B(H) SQSTM1 (H)		12 (16.4)	4.01 (1.38–11.66)	**0.011**	9.49 (2.19–41.08)	**0.003**

Abbreviations: SCC, squamous cell carcinoma; CHR, crude hazard ratio; CI, confidence interval; AHR, adjusted hazard ratio. * *p*-value were adjusted for cell differentiation (moderate + poor vs. well) and AJCC pathological stage (stage III + IV vs stage I + II) by multiple Cox’s regression. Bold values denote statistically significant.

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
