# Peer review of "Map1lc3b and Sqstm1 Modulated Autophagy for Tumorigenesis and Prognosis in Certain Subsites of Oral Squamous Cell Carcinoma"

_jcm, 2018, doi:10.3390/jcm7120478_

Round 1
Reviewer 1 Report
In this study, authors examined the role of autophagy in buccal mucosal SCC (BMSCC), tongue SCC (TSCC), and lip SCC (LSCC). Authors found that the expression levels of autophagy marker, MAPLC3B and cytoplasmic SQSTM1 were elevated in the tumor tissues of three subsites compared with those in adjacent normal tissues by using tissue microarray comprised 498 OSCC patients, including 181 BMSCC, 244 TSCC, and 73 LSCC patients. Moreover, they found that the coexpression of higher MAPLC3B and SQSTM1 demonstrated a significantly worse DSS and DFS in patients with BMSCC and LSCC, but not TSCC. In vitro experiments revealed that knockdown of MAPLC3B and SQSTM1 reduced autophagy, cell proliferation, invasion and tumorspheres in BMSCC cells. In addition, knockdown of both MAPLC3B and SQSTM1 enhanced the cytotoxic effects of paclitaxel in the tumorspheres of BMSCC cells. Authors suggest that MAPLC3B and SQSTM1 may modulate autophagy to facilitate tumorigenesis and chemoresistance in OSCC, particular in BMSCC.
I feel that this paper contains interesting findings. However, I feel that the results in this study are limited and too preliminary. There are several questions as the following;
1. Immunohistochemical findings showed the difference among the lesion such as BMSCC, TSCC and LSCC. For example, MAPLC3B was associated with a poor prognosis only in TSCC; the coexpression of higher MAPLC3B and SQSTM1 demonstrated a significantly worse DSS and DFS in patients with BMSCC and LSCC, but not TSCC. How do authors explain these?
2. In in vitroexperiments, authors used only BMSCC cells. Authors should use TSCC and LSCC for in vitroexperiments to compare the phenotypes.
3. Authors found that knockdown of MAPLC3B and SQSTM1 reduced autophagy, cell proliferation, invasion and tumorspheres in BMSCC cells. How do authors think about the correlation between reduced autophagy and proliferation, invasion and tumorspheres?
Author Response
Please find attached file for the detailed response and additional results. Thank you very much!

Reviewer 2 Report
This is a well written manuscript with good study design, appropriate statistical analyses, and relevance to the field. I only have minor comments related to how the information in presented. The authors occasionally use LC3 and P62 instead of MAPLC3B and SQSTM1. This is confusing to the reader. Also in figures 6A and B, the authors label the lanes C, L, and S. I am guessing that C is control, L is MAPLC3B, and S is SQSTM1. Some clarification will make it easier for readers to understand. Otherwise, this is a very well written manuscript.
Author Response
Thank you so much for the great comments. The English writing has been edited by American Journal Experts". Please also find attached file for the detailed response.

Round 2
Reviewer 1 Report
Authors responded there reviewers' comments. Therefore, I think that this paper is acceptable for publication.